# Unravelling some factors affecting sexual reproduction in rock-specialist shrub: Insight from an endemic *Daphne arbuscula* (Thymelaeaceae)

**Zuzana Gajdošová**[1], **Marek Šlenker**[1], **Marek Svitok**[2,3], **Gabriela Šrámková**[4], **Drahoš Blanár**[5], **Veronika Cetlová**[1], **Jaromír Kučera**[1], **Ingrid Turisová**[6], **Peter Turis**[6], **Marek Slovák**[1,4]*

1 Institute of Botany, Slovak Academy of Sciences, Bratislava, Slovak Republic, 2 Department of Biology and General Ecology, Technical University in Zvolen, Zvolen, Slovak Republic, 3 Department of Forest Ecology, Czech University of Life Sciences Prague, Suchdol, Praha, Czech Republic, 4 Department of Botany, Charles University, Praha, Czech Republic, 5 Muránska planina National Park Administration, Muráň, Slovak Republic, 6 Department of Biology, Ecology and Environment, Matej Bel University in Banská Bystrica, Banská Bystrica, Slovak Republic

* marek.slovak@savba.sk

**Data Availability Statement:** The data are provided as Supporting information. The datasets generated for this study can be found in online repositories

## Abstract

The role of endemic species in global biodiversity is pivotal, and understanding their biology and ecology is imperative for their fitness and long-term survival, particularly in the face of ongoing climatic oscillations. Our primary goal was to investigate the sexual reproduction level of the endangered Western Carpathian endemic *Daphne arbuscula* (Thymelaeaceae), which inhabits extreme rocky habitats, and to comprehend the influence of specific factors on its reproductive success. We conducted the research across four populations, varying in size and environmental conditions. Over two years, we monitored flower and fruit production, analyzed genetic variability within and among populations, and studied pollination mechanisms. *Daphne arbuscula* proved to be strictly self-incompatible, with significant variations in flower and fruit production among populations and seasons. The average fruit production percentage consistently remained below 50% across populations, indicating challenges in sexual reproduction. Cold and harsh weather during the reproductive phase had a substantial negative impact on sexual reproduction efficacy, leading to decreased fruit production. Nevertheless, several individuals in sheltered microhabitats displayed significantly higher fruit production, ranging from 60% to 83%, emphasizing the critical role of microhabitat heterogeneity in sustaining sexual reproduction in this species. We found no pronounced differences in genetic diversity within or among populations, suggesting that genetic factors may not critically influence the reproductive success of this endemic species. The implications of our findings might be of paramount importance for the long-term survival of *D. arbuscula* and offer valuable insights for the development of effective conservation strategies for this species.

(https://www.ncbi.nlm.nih.gov/genbank/) with GenBank accession numbers OQ269432–OQ269447 for 16 ITS sequences original for this study and OQ632972–OQ632991 for 20 cpDNA sequences (rpl32–trnL region). The RADseq reads have been deposited in the NCBI Short Reads Archive (BioProject ID PRJNA989661, SRA: SRR25100387–SRR25100425).

**Funding:** This work was supported by the Slovak Research and Development Agency (no. APVV-22-0365 to MS); by Scientific Grant Agency of the Ministry of Education, Science, Research and Sports of the Slovak Republic and the Slovak Academy of Sciences (no. VEGA 2/0098/22 to JK) and 'Conserving the endemic flora of the Carpathian Region', (to MS and JK) managed by the Institute of Botany of the Slovak Academy of Sciences. Computational resources were provided by the e-INFRA CZ project (ID: 90254), supported by the Ministry of Education, Youth, and Sports of the Czech Republic. M. Svitok was supported by the Operational Programme Integrated Infrastructure (OPII), funded by the ERDF (ITMS 313011T721). Funders did not play any role in study design, data collection and analysis, decision to publish, or preparation of the manuscript.

**Competing interests:** He authors have declared that no competing interests exist.

## Introduction

Endemism is widely recognized as a key factor in prioritising biodiversity conservation strategies [1, 2]. To ensure the effective protection of endemic plants, it is crucial to have a comprehensive understanding of their evolution, life history, biology, and ecology and to identify the factors and mechanisms that contribute to their fitness and long-term survival [1–7]. In this regard, understanding the limitations that hinder their colonization of broader distribution areas is of particular interest [6, 8–11]. Furthermore, numerous studies have demonstrated that endemic species often exhibit lower reproductive success or dispersal abilities compared to their widespread congeners [3, 4, 9, 11]. Endemics are frequently habitat specialists that have evolved more specific reproductive and pollination biology [11]. A decrease in lower seed production, for example, may be restricted by diminished pollen investment [4, 9], pollinator deficiency [12], or increased seed predation [13]. The mode of reproduction also has a significant impact on the genetic structure of populations, particularly in species with small and isolated populations [10, 11, 14, 15]. Additionally, the presence of selfing or increased clonality might lead to decreased genetic diversity within and among populations, genetic drift, and inbreeding depression [3, 6, 15–17]. However, dispersal ability might be strongly influenced by environmental conditions, and the phylogenetic context of the studied plants should also be taken into consideration [4, 6]. Nevertheless, it is crucial to bear in mind that not all endemic plant species face reproductive and dispersal challenges when generalizing these findings [3, 6].

This study focuses on *Daphne arbuscula* Čelak. (Thymelaeaceae), a narrow endemic shrub occurring in the Muránska planina Mts in the Western Carpathians, Slovakia [18–20]. It serves as an ideal model for investigating sexual reproduction in endemics due to its unique ecological characteristics and habitat preferences. It is referred to as an outcrosser and is predominantly pollinated by various insect species [21, 22]. The fruit production is reported to be low, and this could potentially have a negative impact on its fitness and survival [cf. 19, 21, 23]. *Daphne arbuscula* is primarily a saxicolous plant that thrives in extreme habitats, particularly on relict limestone cliffs and rocks in the mid-altitudinal montane zone (S1 Fig in S1 File) [18–20]. Although these habitats provide a diverse range of microclimatic conditions, most of them are susceptible to climatic fluctuations. Compared to grassland or forest vegetation, open rocky habitats are more vulnerable to oscillations in temperature, droughts, a lack of protective snow cover, limited root space, intense UV radiation, wind exposure, and gravitational challenges [24–27]. Such extreme environmental conditions experienced during the reproductive phase of plants can negatively affect the formation and development of sexual organs and reproductive propagules, potentially leading to impaired sexual reproduction [28–32].

The populations of *D. arbuscula*, although limited to one mountain system, demonstrate significant niche diversity, specifically those residing at lower altitudes and facing south have typically drier and warmer climatic conditions, whereas those living at higher altitudes on the northern slopes experience harsher and wetter conditions with a montane microclimate [19, 20]. All populations are naturally isolated by dense mixed forest vegetation, resulting in a forest-free 'island system' that is essentially unconnected [cf. 24]. These ecological configurations might have significant implications for the reproductive success of the species and can influence the genetic structure within and among populations. Furthermore, in certain cases, smaller populations with fewer individuals may deviate from major reproductive systems and favour autogamy or vegetative reproduction. Such shifts can significantly influence genetic variability, fitness, and the survival of the given species [33, 34]. Previous studies have provided incomplete data on the biology of *D. arbuscula*, including its reproductive system [19, 21, 22, 35–39]. In addition, these studies were conducted with small sample sizes and did not consider

environmental variations among populations. As a result, our understanding of its reproductive strategies, fitness, and survival abilities remains limited.

The primary objective of this study is to explore the overall fruit production of *D. arbuscula* and test whether it is decreased as previously hypothesised [19, 21, 23]. Furthermore, the study aims to investigate the correlations between flower and fruit production and seasonal climatic patterns throughout the reproductive phase over a span of two years. Additionally, we will examine whether *D. arbuscula* exhibits strict self-incompatibility or if there are any indications of a transition towards some rarer sexual reproduction mechanism. It also entails analysing the genetic variability within and among populations, considering the size and ecological traits of the populations under examination.

## Material and methods

### Study species

*Daphne arbuscula* is a long-lived, evergreen shrub that grows to a height of 10–30 cm, has coriaceous leaves, and bears bisexual, strongly fragrant flowers ranging in colour from pink to purple, rarely even white (S1 Fig in S1 File) [18–20, 40]. The species is diploid, with a chromosome count of 2n = 2x = 18 [35, 39, 41]. Fruits are one-seeded hard dry or fleshy drupes in brownish to yellow-green colours, often enclosed within marcescent hypanthium structures (S1 Fig in S1 File) [19, 38].

### Sampling and design of the study

Four localities were selected to represent the entire range of ecological conditions within the whole distribution range of *D. arbuscula* in Muránska planina Mts (Fig 1) [19]. Moreover, the presence of clonal growth in *D. arbuscula* [19] made it difficult to determine the exact number of genetically distinct individuals. To reduce bias in the final results and the interpretation of flower and fruit production, only individuals spatially separated by smaller rock formations or vegetation were considered, minimizing the chances of clone inclusion. Regarding population size, Poludnica is the largest locality, with over 1400 shrubs, followed by Veľká Stožka, with approximately 1100 plants, then Malá Stožka with around 500 shrubs, and lastly, Šiance, the smallest locality with about 115 shrubs. The precise estimation of shrub numbers is challenging due to the highly inaccessible rocky terrain, making precise counting essentially impossible. The study sites, Poludnica and Šiance, predominantly face south and southeast and are situated at lower altitudes (approximately 800 m and 950 m above sea level). They represent a niche characterized by drier and warmer climatic conditions and are hence referred to as 'warm-microclimate' localities. The Šiance (SI; Fig 1; S1 Fig in S1 File) population is limited to small calcareous rock bodies, with most individuals located in open, sunny rocks (SI41–SI46). A few individuals (SI47–SI50) occur in shaded rock crevices beneath a mixed forest canopy on northern-oriented rocks. In contrast, the other pair of populations, Malá Stožka and Veľká Stožka, represent a niche with colder and more humid climatic conditions, predominantly facing north, and are located at higher altitudes (approximately 1150 and 1275 m above sea level). These localities are hence referred to as 'cold-microclimate' sites.

Ten mature individuals of approximately the same size were selected from each population and marked with a unique metal identifier for subsequent identification purposes (S1 Table in S1 File). All the examined populations were of sufficient size to ensure that the research did not negatively impact their fitness and viability. Population size was estimated using data from the monitoring of *D. arbuscula* conducted by the Muránska planina National Park Administration. Authorization for field research and the collection of *D. arbuscula* samples, designated

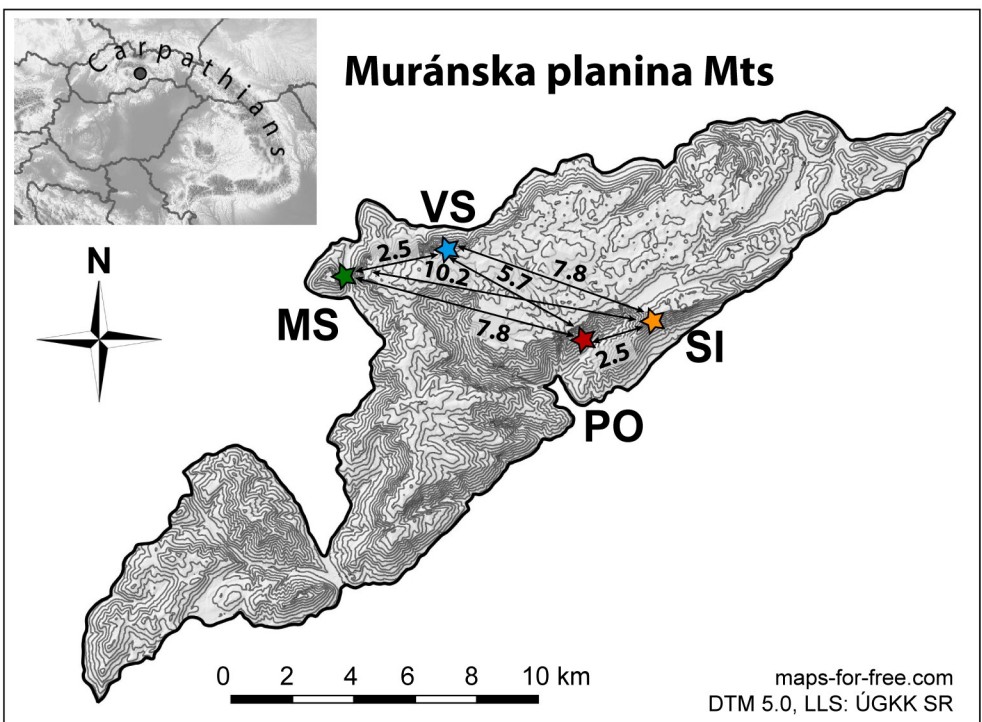

**Fig 1. Distribution map of studied populations of *Daphne arbuscula* in the area Muránska planina Mts in the Western Carpathians and distances among the studied populations.** Population codes are as follows: MS–Malá Stožka, PO–Poludnica, SI– Šiance, VS–Veľká Stožka.

by permit number 5408/2017-6.3, was granted by the Ministry of Environment of the Slovak Republic on July 12, 2017. This permission facilitated the collection of all analyzed materials from the designated study locations throughout the entire investigative period, spanning from 2017 to 2019.

## Flower and fruit production

The fruits typically ripen within 4–5 weeks; however, this process is highly individual-specific and influenced by climatic conditions. Subsequently, the fruits disperse rapidly and haphazardly from shrubs within a few days. Therefore, after the majority of the flowering shoots on the studied individuals had been in the full-bloom stage for at least a few days, securing pollination, they were isolated using nylon bags to ensure precise analysis of the fruit set.

For each of the ten selected individuals in all populations, fifteen flowering shoots with one inflorescence were randomly chosen and isolated using nylon bags with a mesh size of approximately 0.5 mm (S1 Fig in S1 File). A total of 150 shoots per population were isolated during each season. The flowering period of this species is relatively short, spanning two to three weeks in May and/or June. Populations from cold-microclimate localities often exhibit delayed flowering, sometimes up to one month compared to populations from warm-microclimate sites [20, 23, 36]. The isolation process therefore took place in May and June, depending on local conditions (S1 Fig in S1 File). The number of flowers in each inflorescence was recorded, and the fruits were collected between June and July, once they had ripened and started dropping from the shoots. The number of fruits formed per branch was recorded. The study was conducted over two growing seasons, from April to July in both 2018 and 2019.

Previous studies have identified fruit polymorphism, which includes two distinct fruit morphotypes that differ in colour and mesocarp thickness and are dispersed at approximately the same time [23, 38]. However, it remains uncertain whether these morphotypes represent true fruit polymorphism or different ontogenetic stages of the fruit ripening process. Despite seeds from the 'small-fruit' morphotype appearing to have lower viability compared to those from the 'big-fruit' morphotype, their viability still exceeds a threshold of 50% [38]. This means that both morphotypes might contribute significantly to the sexual reproduction of this species, and therefore we decided not to differentiate the collected fruits by morphotype in this study.

Generalized linear mixed models (GLMM) [42] were utilized to investigate temporal and spatial differences in flower and fruit production of *D. arbuscula*. The models incorporated random effects of repeatedly measured individuals and fixed effects of time (years 2018 and 2019), population (Šiance, Malá Stožka, Veľká Stožka, and Poludnica), and their interaction. Initially, the number of flowers was fitted using a GLMM with a Poisson error distribution and a logarithmic link function. However, the model residuals displayed considerable under-dispersion ($\varphi = 0.28$), which can lead to overestimated standard errors and misleading statistical inference [43]. To address this issue and minimize the type II error rate, we refitted the data using a flexible Conway-Maxwell-Poisson distribution, allowing us to model a wide range of dispersion levels [44]. No further violation of the model assumptions was detected in diagnostic plots of residuals.

The probability of fructification was calculated as a proportion of flowers that developed fruits out of all flowers recorded on a given plant. Initially, we modelled the data as binomial proportions with a logit link function. However, the binomial GLMM showed significant over-dispersion ($\varphi = 1.33$), potentially leading to inflated type I error rates [45]. As a result, we refitted the data using a model with beta-binomial distribution [46] and a complementary log-log link function, efficiently handling the overdispersion. Likelihood ratio tests were employed to assess the significance of the model terms [42]. Whenever a significant overall effect was identified ($\alpha = 5\%$), we conducted a series of post hoc tests with Tukey-adjusted probabilities to compare the estimated marginal means [47]. Marginal pseudo-determination coefficients were calculated for each GLMM to quantify the proportion of the total variation explained by the random and fixed effects (R2c) and fixed effects only (R2m) [48].

The analyses were performed in R v. 4.2.0 [49] using the libraries DHARMa [50], emmeans [51], ggplot2 [52], glmmTMB [53] and performance [54].

## Breeding system study

To test for autogamy, nylon bags with a smaller mesh size of less than 0.2 mm were utilized to prevent pollinators from transferring pollen from other inflorescences or plants. A preliminary study (S2 File) showed that a mesh size of 0.5 mm was not effective at preventing pollination by thrips (Thysanoptera), known to be important pollinators of *D. arbuscula* [17, 24]. During the 2018 growing season, three inflorescences with unopened buds were isolated from each individual in all populations, totalling 120 shoots. Furthermore, to determine compatibility (SC vs SI), assisted pollination studies were conducted on a subset of plants (3–5 shrubs) at the Poludnica and Šiance localities. After the flowers opened, the nylon bags were removed from the isolated inflorescences, and the flowers were manually pollinated between flowers within the same inflorescence using a delicate, hand-made brush. Following pollination, the flowers were enclosed in nylon bags to prevent pollen transfer between flowers or from other plants via animal vectors. Bags from all studies were collected once the fruits ripened.

## Genetic analyses

We conducted a comprehensive genetic study on a species that lacked previous genetic data. Thus, to obtain the first insight into the genetic variation of the studied species, we employed traditional molecular markers, specifically the ITS1-5.8S-ITS2 region of ribosomal DNA (referred to as ITS) and the *rpl32-trnL*$^{(UAG)}$ region of chloroplast DNA (S3 File). For a more detailed investigation of population-level variability, we utilized double-digest RAD sequencing (dd RADseq; all 40 individuals).

Genomic DNA was extracted from young dried leaves of all sampled individuals using the Invisorb® Spin Plant Mini Kit (Invitek Molecular GmbH, Berlin, Germany) following the manufacturer's instructions, supplemented with PVP and 6 μL of RNase A. The extracted DNA was then purified using AMPure XP (Beckman Coulter Inc., Brea, California, USA).

**dd RAD sequencing.** All 40 individuals from four populations were genotyped for single nucleotide polymorphisms (SNPs) using the dd RADseq protocol of [55] with modifications [see 56]. The libraries were sequenced with 300 cycles (2×150 bp paired-end/PE reads) using the Illumina HiSeq platform (Macrogen Inc., Seoul, South Korea).

**Raw RADseq data processing, variant calling, and filtration.** The raw data processing, variant calling, and filtration were conducted as follows: the raw reads were demultiplexed using FASTX-Toolkit v. 0.0.14 [57], quality trimmed in Trimmomatic v. 0.36 [58], and filtered for PCR clones using the clumpify.sh script in BBTools v. 38.42 [59]. Since no reference genome was available, a de novo catalogue of RADseq loci was constructed using STACKS v. 2.53 [60, 61]. To determine the optimal Stacks parameters specific to our dataset and minimize erroneous splitting or lumping of loci, we followed the methods outlined in [62, 63]. The final catalogue of RADseq loci was built using the following criteria: a minimum coverage to identify a stack of 20× (-m 20), a maximum number of mismatches allowed between loci in each sample (-M) of 2, and a maximum number of differences among loci to be considered as orthologous (-n) of 2. The cluster_fast command (USEARCH v. 11; [64]) was then utilized to combine similar sequences with more than 80% identity to create the non-redundant clusters. Subsequently, mapping on the non-redundant de novo catalogue of RADseq loci was performed using BWA v. 0.7.5a [65] and processed with Picard Tools v. 2.22.1 [66]. The GATK v. 3.7 [67] was employed to call and filter reliable variants. Only biallelic SNPs were selected, but variants matching the following criteria were removed: Quality by Depth (QD) < 2.0, FisherStrand (FS) > 60.0, RMS Mapping Quality (MQ) < 40.0, MQRankSum < -12.5, ReadPosRankSum < -8.0, SOR > 3.0. Furthermore, genotypes with a read depth lower than 8, samples with more than 40% missing genotypes, and variants with more than 30% missing genotypes were discarded.

**RADseq data processing and analyses.** To reveal between and within-population variability of the studied populations, multiple approaches were employed. Principal component analysis (PCA) was conducted using the adegenet package [68] in R 4.2.0 software [49]. To assess the potential genomic admixture of *D. arbuscula* individuals, the assignment of individuals to homogeneous clusters was inferred with ADMIXTURE 1.3 [69]. To mitigate linkage disequilibrium while considering the entire dataset, 100 datasets were generated by randomly selecting one SNP from each RAD locus. Subsequently, SNP data were further filtered by removing sites with a minor allele frequency <5.0% to reduce bias associated with erroneous genotypes and singletons [70]. For each dataset and K values ranging from 1 to 4, one replicate of ADMIXTURE was run using the AdmixPipe pipeline [71]. Results were summarized by Clumpak [72] with a similarity threshold of 0.9 for major clusters, aiding in the identification of potentially discordant modes within a single K value. To visualize potential conflicting signals, a phylogenomic network was inferred using the neighbour-net algorithm in SplitsTree software [73] based on the Nei's genetic distance matrix calculated in the StAMPP R package [74].

A total of 8,394,840.59 (±3,588,224.75 SD) high-quality RADseq reads were generated per sample, except for sample SI41, which yielded 533,938 reads. A de novo catalogue of RADseq loci was constructed using STACKS, resulting in 23,494 loci. After combining similar sequences with over 80% identity, the non-redundant catalogue comprised 17,485 loci. On average, 61.30% (±1.31% SD) of the reads were successfully mapped to this catalogue, resulting in a final mean coverage of 340.28× (±145.8 SD) for all samples, except SI41, which had a mean coverage of 20.6×. The raw dataset contained 38,368 SNPs. The sample SI41 was excluded from the final analysis due to high levels of missing data. Thus the final dataset consisted of 39 individuals. The RAD-seq reads have been deposited in the NCBI Short Reads Archive under the BioProject ID PRJNA989661 and SRA accession number SRR25100388-SRR25100425.

## Results

### Interpopulation, intrapopulation, and seasonal variability in flower and fruit production

Flower production of *D. arbuscula* significantly varied among populations, although the differences were contingent upon the year of observation (time × population: $\chi^2 = 41.8$, df = 3, $p < 0.0001$). In 2018, the number of flowers produced across populations was relatively consistent (Fig 2A). However, the following year, a decline in flower production was observed across all populations, with the most significant reduction occurring in Malá Stožka, which diverged significantly from the flower number of Poludnica (t = 3.5, df = 1166, p = 0.0029) and Veľká Stožka population (t = 4.2, df = 1166, p = 0.0002). Although the overall model was highly significant ($\chi^2 = 508$, df = 8, $p < 0.0001$), the flower counts data were fitted relatively poorly (R2c = 0.09, R2m = 0.04).

Fruit development took between 5 and 7 weeks after anthesis, depending on microclimatic conditions. The fruit production was low, ranging from 8.97% to 48.49% across populations and both growing seasons (Table 1; Fig 2B). The populations showed significant differences in fruit production between the years (time × population: $\chi^2 = 258$, df = 3, $p < 0.0001$). In 2018, the fruit set was significantly higher in the warm-microclimate locality Šiance than in the cold-

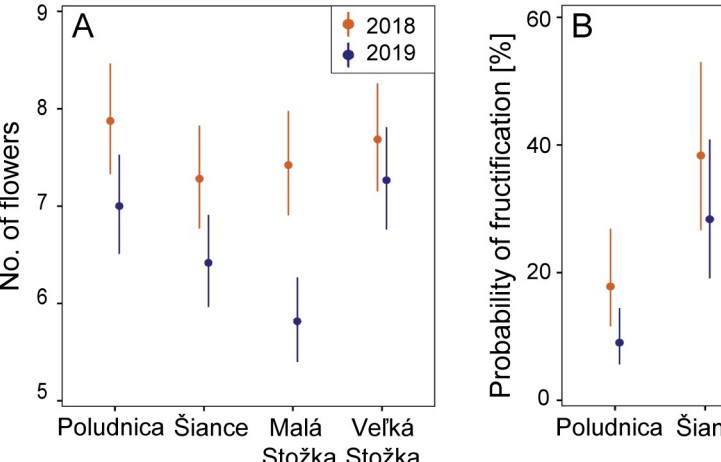

**Fig 2. Seasonal variability in sexual reproduction of the four studied populations of *Daphne arbuscula*.** (A) the number of flowers in the inflorescence; (B) the probability of fructification. The estimates based on Generalized Linear Mixed Models (GLMM) are represented by circles, with their 95% confidence intervals displayed as error bars.

**Table 1. Reproduction characteristics of the studied populations of *Daphne arbuscula*.**

| Population | Poludnica | Šiance | Malá Stožka | Veľká Stožka |
|---|---|---|---|---|
| Number of studied individuals | 10 | 10 | 10 | 10 |
| Population size estimate | 1400 | 115 | 500 | 1100 |
| **Flower and fruit production** | | | | |
| 2018 | | | | |
| Flower production | 1187 | 1100 | 1121 | 1159 |
| Fruit production | 241 | 471 | 118 | 104 |
| Fruit set [%] | 20.30 (0–43.1) | 42.82 (17.9–76.1) | 10.53 (2.4–30.9) | 8.97 (0–19.4) |
| 2019 | | | | |
| Flower production | 1055 | 909 | 794 | 1096 |
| Fruit production | 115 | 263 | 385 | 390 |
| Fruit set [%] | 10.90 (0–33.0) | 28.93 (2.5–83.3) | 48.49 (24.3–70.1) | 35.58 (12.4–66.7) |

microclimate populations of Malá Stožka (t = 4.54, df = 1166, p < 0.0001) and Veľká Stožka (t = 5.61, df = 1166, p < 0.0001; Fig 2B). Poludnica population also showed a significantly higher fruit set than Veľká Stožka (t = 2.95, df = 1166, p = 0.017). However, in 2019, the opposite was true; the warm-microclimate population of Poludnica showed significantly lower fruit production than Šiance (t = 3.67, df = 1166, p = 0.0014) and the cold-microclimate populations of Malá Stožka (t = 5.35, df = 1166, p < 0.0001) and Veľká Stožka (t = 4.20, df = 1166, p = 0.0002). In Poludnica and Šiance fruit production was approximately equal in both years, and the temporal pattern seems to match flower counts. In cold climate populations, considerable variability was observed in fruit production between the two years, which was less affected by flower production. The beta-binomial GLMM was highly significant ($\chi2$ = 508, df = 8, p < 0.0001) and had good performance ($R2c$ = 0.53, $R2m$ = 0.30).

The success rate of sexual reproduction at the individual level fluctuated more dramatically, ranging from 0% to 83% (S1 Table in S1 File). A few individuals did not produce any fruits during the study period (e.g., PO16). In contrast, only a few individuals from populations Šiance, Malá Stožka, and a single individual from Veľká Stožka exhibited fruit production in excess of 50% (S1 Table in S1 File). During both seasons (S1 Table in S1 File), certain individuals from locality Šiance, specifically SI47–SI49, exhibited remarkably high rates of fruit production (60–83%).

## Breeding system study

During the test for autogamy, no fruits developed from 120 inflorescences. Fruit production was not observed in any of the isolated inflorescences, even in those with assisted self-pollination to determine compatibility (Table 2).

**Table 2. Breeding system testing in studied populations of *Daphne arbuscula*.**

| Population | No. of bagged flowers | No. of hand-pollinated flowers | No. of fruits formed |
|---|---|---|---|
| Poludnica | 228 | 87 | 0 |
| Šiance | 214 | 81 | 0 |
| Malá Stožka | 216 | 0 | 0 |
| Veľká Stožka | 227 | 0 | 0 |

## Genetic diversity

The genomic RADseq analyses consistently supported the presence of shallow population structure, indicating genetic distinctiveness among all four populations. PCA and ADMIXTURE analysis at K = 2 and K = 3 showed that Poludnica and Veľká Stožka populations are genetically closer, with separation only occurring at K = 4 (Fig 3A and 3B). Interestingly, even at K = 4, a minor cluster revealed their genetic similarity. Sample VS29 is likely in an outlier position in Splitstree analysis because it has genes from two different gene pools (most likely populations Poludnica and Veľká Stožka). In contrast, Malá Stožka was identified as the most divergent population among all the populations analyzed. Both ADMIXTURE analysis and the phylogenomic network generated by the neighbour-net algorithm suggested limited gene flow between the analyzed populations (Fig 3A and 3C). The four populations exhibited different numbers of private alleles, ranging from 1323 (Šiance– 3.4% of total alleles) to 1958 (Poludnica– 5.1% of total alleles; S2 Table in S1 File). No private fixed alleles were detected in the population. Further population-genetic parameters showed only negligible differences (S2 Table in S1 File). The low levels of between-population divergence were further supported by the pairwise population differentiation values ($F_{ST}$), which ranged from 0.0035 to 0.0079 (Table 3).

## Discussion

### Seasonal and population fluctuations in *D. arbuscula* fruit formation

Endemic species with restricted distribution ranges and small population sizes often exhibit decreased ability of sexual reproduction, especially when compared to their more widespread

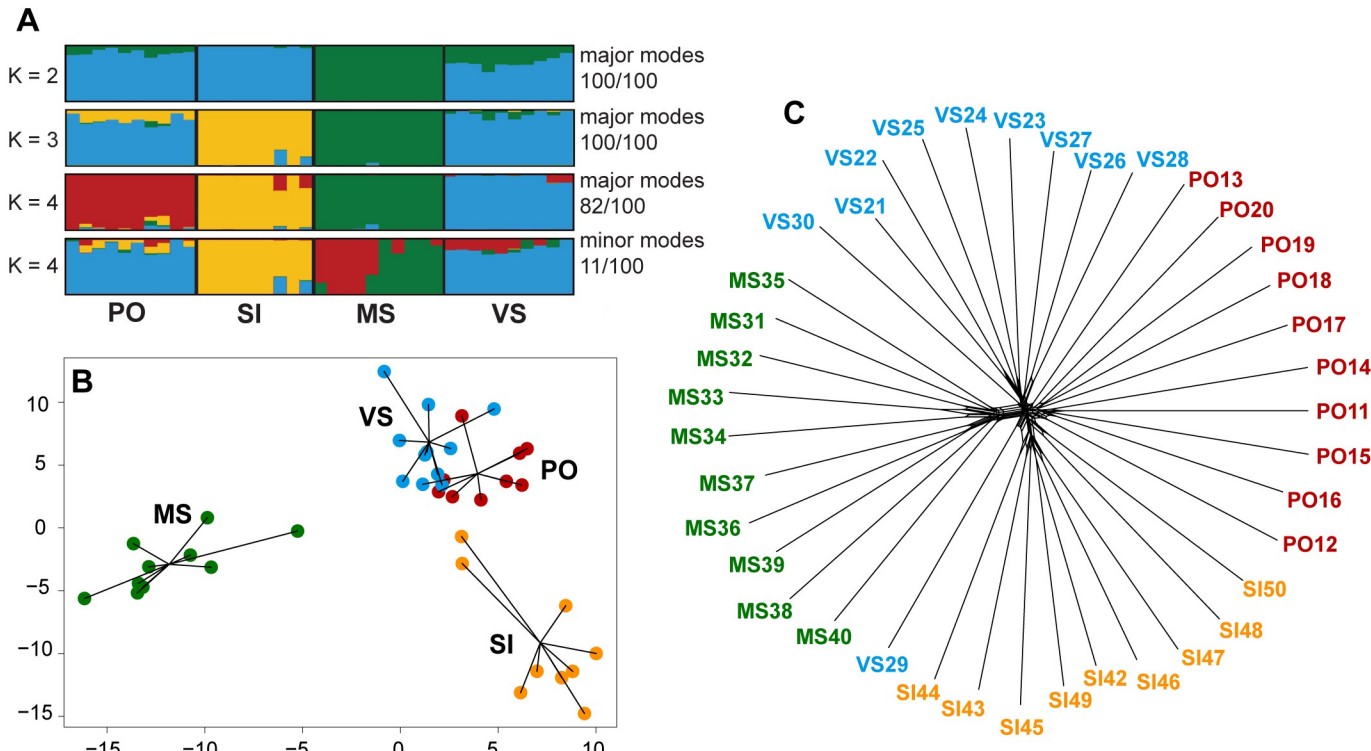

**Fig 3. Population genetic analyses based on 38,368 single nucleotide polymorphisms and 39 individuals from four populations of *Daphne arbuscula*.** (A) genetic clustering depicted by ADMIXTURE analysis at K = 2–4. Each individual is represented by a vertical bar, with colour proportionally indicating the segment assignment; (B) Principal component analysis (PCA) analysis. (C) Neighbour-Net as recovered by Splitstree. Population codes are as follows: MS–Malá Stožka, PO–Poludnica, SI– Šiance, VS–Veľká Stožka.

**Table 3. Pairwise population $F_{ST}$ estimates inferred from single nucleotide polymorphism data.**

|  | Poludnica | Šiance | Malá Stožka | Veľká Stožka |
|---|---|---|---|---|
| **Poludnica** | – | 0.0042 | 0.0063 | 0.0035 |
| **Šiance** | – | – | 0.0079 | 0.0055 |
| **Malá Stožka** | – | – | – | 0.0058 |

congeners [3, 4, 8–11, 13, 23, 75]. This may also limit their dispersal abilities and threaten their survival, in particular in the case of various disturbances. Reproduction effort can be influenced by a whole range of internal and external factors, e.g. breeding system, pollinator activity or climatic conditions.

Our study revealed that overall sexual reproduction in endemic *D. arbuscula* also exhibited relatively low fruit production, not exceeding 50% at the population level. We presented evidence that climatic conditions during the flowering and fruiting periods exert a great influence on fruit production, surpassing factors such as genetic variability, location, and ecology of the studied populations [cf. 76, 77]. Seasonal climatic conditions were also identified as a key determinant of sexual reproduction success in high-mountain cushion saxicolous members of the genus *Saxifraga* [78, 79]. In general, it has been demonstrated that unfavourable climatic conditions during the reproduction stage have the potential to impact plant reproduction, with species exhibiting diverse responses [28, 31, 80].

*Daphne arbuscula* flowers from late April to June, coinciding with substantial weather variability in Central European mountainous regions [81]. In 2018, drier and warmer weather in April and May, followed by colder and wetter conditions in June, resulted in significantly higher fruit production in the earlier-flowering warm-microclimate populations (20.3% in Poludnica; 42.82% in Šiance; Table 1) compared to the later-flowering cold-microclimate counterparts (10.53% in Malá Stožka; 8.97% in Veľká Stožka; Table 1). Conversely, in 2019, colder and wetter weather in April and May led to a remarkable decrease in fruit formation in warm-microclimate populations (10.9% in Poludnica; 28.93% in Šiance; Table 1), while drier and warmer conditions in June enhanced fruiting in cold-microclimate populations (48.49% in Malá Stožka; 35.58% in Veľká Stožka; Table 1). These findings indicate that cold and wet weather during the reproductive phase diminishes fruit production. However, it appears that fruit production in populations residing in warmer environments demonstrated greater stability across both seasons, suggesting that they were less affected by climatic fluctuations compared to populations with a cold microclimate (Fig 2B). This indicates that the sexual reproduction behaviour of populations residing in cold microclimates at north-oriented sites shows an increased vulnerability to between-year oscillations in a seasonal climate, which could exacerbate their susceptibility to ongoing trends in global climatic changes. Nevertheless, it is reasonable to approach our findings with caution due to some limitations in our study design. The lack of replication for each location, representing distinct climate types and population sizes, renders the design insufficiently robust, potentially strengthening the investigation's results. Adding more replications, however, presented considerable challenges, given the difficulty or inaccessibility of the majority of populations of this endemic shrub located on rocky cliffs with low population sizes. Similarly, the study's limitation to two seasons may be considered a constraint in drawing definitive conclusions. While we acknowledge that extending the study period to 10 or 15 years would provide sufficient data to discern clear trends in the impact of seasonal climatic conditions on fruit production, such an extended duration is evidently unfeasible for the majority of scientific projects today. Nevertheless, our observations across two seasons, encompassing climatically disparate years with divergent patterns during

the reproductive phase of this species (Fig 2B and S1 File), highlight the influence of seasonal climate conditions on the fruit production of the studied species, even at the microscale level. Despite these constraints, we believe that these limitations have not unduly biased our outcomes or, particularly, the interpretation of our data.

Harsh climatic conditions, such as cold, high humidity, and strong winds, can damage gametes, flowers, and young fruits, leading to lower seed quality and fruit set [29, 31, 32]. [37] found that late frosts, prolonged droughts, and strong winds during fruit formation can cause tissue degeneration in *D. arbuscula* fruits. The degree of fructification was found to depend on the temperature and occurrence of rather sunny and dry weather in the spring also in glacial relict *Pulsatilla vernalis* [76]. As climate change increases the incidence of extreme climatic events, higher spring temperatures may result in earlier flowering, increasing the risk of exposure to late frosts and subsequent fruit damage [28, 30, 32]. Indeed, late summer frosts exert a more pronounced negative impact on reproductive shoots, consequently affecting sexual reproduction success, compared to vegetative ones in the majority of studied high-mountain plants from the European Alps [82].

Moreover, the flowering period of *D. arbuscula* shrubs is rather short and typically lasts up to two weeks [20, 36]. Previous field studies on *D. arbuscula* have identified potential insect pollinators (i.e., from orders Thysanoptera, Coleoptera, Lepidoptera, Hymenoptera–Formicoidea or Apoidea [22]) in the vicinity of blooming shrubs under sunny and warm weather conditions, but contrasting trends may emerge when unfavourable climatic conditions prevail [19, 21, 22]. Thus prolonged rainfall and low temperatures can further diminish insect pollinator activity, reducing the chances of successful pollination [19, 31, 83]. An additional critical factor adversely impacting successful pollination could be the mismatch in plant phenology and pollinator activity resulting from climate change [e.g. 84]. Such a deficiency in pollinators consistently leads to decreased fruit production in other *Daphne* species as well [85–87].

Nevertheless, rocky habitats exhibit high geomorphological heterogeneity, featuring diverse microsites characterized by distinct microclimates [88–90]. Within *D. arbuscula* populations, individual shrubs located just a few meters apart can significantly vary in flowering time and reproductive success due to microclimate variations. Microsites with hospitable microclimates might act as a buffer against harsh climatic fluctuations, ensuring that some shrubs experience optimal conditions for sexual reproduction. This was clearly demonstrated in the Šiance population, where shrubs in open, sunny rock crevices (SI41–SI46) flowered 1–3 weeks earlier compared to those on north-oriented, more humid rocks shaded by mixed forests. As a result, shrubs from shaded microhabitats produced substantially more fruit during both seasons (60–83%; S1 Table in S1 File) compared to those from open, sunny places (2.5–59%; S1 Table in S1 File). Similar microsite-dependent sexual reproduction success was observed in the high-alpine species *Saxifraga moschata* [78]. Plants from early snowmelt sites exhibited a higher abundance of sexual reproduction structures compared to those from later snowmelt sites. Such microsites may thus play a crucial role as long-term reservoirs for sexual reproduction and the maintenance of genetic diversity within the species [91].

## The role of additional intrinsic and extrinsic factors in decreasing fruit production of *D. arbuscula*

The observed reduction in sexual reproduction in *D. arbuscula* is in line with the phylogenetic context, as other species within the genus *Daphne* also exhibit insufficient levels of fruit formation below 50% [19, 23, 86, 87, 92–98]. For example, the largely deficient fruit formation was observed also in its two closely related narrow endemic species *D. petraea* [99] and *D. rodriguezii* [86]. Nevertheless, there are exceptions, such as the widespread *Daphne mezereum*, which has

been reported to reach up to 72% fruit production [100]. The limitation in fruit production could be linked to the floral structure of the genus, wherein anthers are situated in the floral tube [40], facilitating the dispersal of numerous pollen grains on the stigma of the same flower (i.e., stigma clogging), thus preventing adhesion of pollen from another plant [86, 88, 95, 96]. Consequently, for outcrossing species like *D. arbuscula*, numerous flowers remain unpollinated, or if pollinated, self-pollinated embryos degenerate during ripening [37, 86, 87, 95, 96]. Despite having high pollen viability (approximately 97%) in outcrossing *D. arbuscula*, indicating no significant issues with pollen development and fertility [35, 39] this morpho-anatomical pollination issue can lead to a decline in the number of embryos resulting from outcrossing pollination.

Another potential explanation for the reduced fruit production in studied species is the limited availability of mating partners within and among populations. Several studies have emphasized the significance of population size and density for the reproductive success of animal-pollinated plants [reviewed by 101]. The scarcity of mating partners is further compounded by self-incompatibility, which has also been demonstrated in *D. arbuscula* (Table 2) [21, 36, 37], leading to the failure of viable seed formation from self-pollinated flowers [9, 17, 102].

Furthermore, the decreased fruit production could be attributed to potential shifts towards vegetative (clonal) reproduction [19, 37, 103], which might also significantly reduce the number of compatible mating partners [cf. 4, 8, 104–106]. However, the role of clonality in this context can only be speculated upon since exploring clonality falls outside the scope of the present study.

Ultimately, our analyses consistently demonstrated a decline in fruit production across all investigated populations, regardless of population size (cf. Table 1; S1 Table in S1 File). This suggests that population numbers, particularly the number of mating partners within a population, may not play a decisive role in the reduction of the species' sexual reproduction. Similarly, a lack of correlation between sexual reproduction-associated traits and population size was identified in *Allysum montanum* from the Jura Mountains in France [107]. Conversely, an opposite trend was detected for various plant species by [15]. Nevertheless, given the limitations in our sampling designs (see Discussion above), we exercise caution in making definitive statements on this matter. Further research is warranted to address this question comprehensively.

## Low association between genetic variability at the population level and sexual reproduction in *D. arbuscula*

Genetic mechanisms may contribute to seed and/or fruit abortion, leading to low fecundity and posing a risk to the long-term survival of plants [e.g., 108–113]. Endemic species, often occurring in small and isolated populations, may be more vulnerable to reduced gene flow within and among populations, potentially experiencing genetic drift or inbreeding depression [10, 11, 15–17, 114, 115].

*Daphne arbuscula* is considered an ancient tertiary relict species that likely survived Pleistocene glaciations on the south-facing slopes of Muránska planina Mts [19, 20]. Our genetic analyses, providing the initial glimpse into the genetic variation of the studied species, did not unveil any intraspecific diversity among the examined populations using traditional markers (ITS and cpDNA; see S3 File). Instead, they only revealed a relatively shallow genetic structure when employing the more robust RADseq method (Fig 3A–3C). This finding, contrasting with the pronounced genetic structure observed in the Menorcan narrow endemic *D. rodriguezii* [116] suggests that the fragmentation of *D. arbuscula*'s distribution range in Muránska planina Mts, or alternatively, colonization from potential microrefugia, occurred more recently in the postglacial period. The genetic differentiation among populations is relatively weak, as indicated by the pairwise population differentiation ($F_{ST}$ values), a low level of

within-population private SNPs (ranging between 3.4% and 5.1% per population), and the absence of private fixed SNPs (see Results section and Tables 1 and 3). Moreover, the stronger genetic differentiation observed in the small populations of Malá Stožka and Šiance, as revealed by ADMIXTURE at K = 2–4 (Fig 3A), may be attributed to genetic drift rather than long-term isolation.

While variations in within-population genetic variability were observed in the studied populations, these differences do not seem to be clearly linked to fruit production. Specifically, the most abundant population, Poludnica, exhibited the highest level of genetic variability, yet reproductive success remained consistently low year after year. In contrast, the population of Malá Stožka, which is spatially the most isolated and the second smallest in size with lower within-population genetic variability, had the highest fruit set during the second year among all studied populations (Table 1; Fig 2B). These findings align with studies that have shown that low genetic diversity does not always result in reduced reproductive success [e.g., 117, 118]. Therefore, it can be concluded that the genetic variability of the analyzed populations does not seem to play a crucial role in the fruit production of this species.

## Conclusions

Our study enhanced the overall understanding of sexual reproduction in the narrow endemic shrub dwelling in extreme rocky habitats. We studied important factors influencing its sexual reproduction and uncovered the crucial impact of climatic fluctuations on the sexual reproduction success of its populations. The species' high sensitivity to unpredictable climatic oscillations, particularly during the spring season, underscores the potential threats it may encounter in the face of climate change. However, we also reveal that the geomorphological heterogeneity of rocky habitats provides diverse microclimatic conditions, which could facilitate sexual reproduction and contribute to the long-term survival of this endemic species. The limited and fragmented distribution range of the studied species does not necessarily lead to significant genetic depletion in the populations. Despite their restricted geographic range, these populations maintain relatively stable genetic diversity, indicating the presence of mechanisms that preserve genetic variation within their habitat. Studying endemic species like *D. arbuscula* provides valuable insights into their survival strategies and the challenges they face. By improving our knowledge of their biology and ecology, we can develop effective conservation measures to mitigate the potential effects of climate change and safeguard the unique components of biodiversity represented by species like *D. arbuscula*. We also present another case of limited sexual reproduction within the genus *Daphne*.

## Supporting information

**S1 File.**
(RAR)

## Acknowledgments

We are deeply thankful to Ivan Bryndza for his assistance in the field.

## Author Contributions

**Conceptualization:** Marek Slovák.

**Data curation:** Zuzana Gajdošová, Marek Šlenker, Marek Svitok, Gabriela Šrámková, Veronika Cetlová.

**Formal analysis:** Zuzana Gajdošová, Marek Šlenker, Marek Svitok, Veronika Cetlová.

**Funding acquisition:** Marek Svitok, Jaromír Kučera, Marek Slovák.

**Investigation:** Zuzana Gajdošová, Drahoš Blanár, Jaromír Kučera, Ingrid Turisová, Peter Turis, Marek Slovák.

**Methodology:** Zuzana Gajdošová, Marek Šlenker, Marek Svitok, Gabriela Šrámková.

**Project administration:** Jaromír Kučera, Ingrid Turisová, Marek Slovák.

**Supervision:** Marek Slovák.

**Writing – original draft:** Zuzana Gajdošová, Marek Slovák.

**Writing – review & editing:** Zuzana Gajdošová, Marek Šlenker, Marek Svitok, Gabriela Šrámková, Drahoš Blanár, Veronika Cetlová, Jaromír Kučera, Ingrid Turisová, Peter Turis, Marek Slovák.

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
