## [Decision Letter · Decision Letter 0]

7 Nov 2023

PONE-D-23-32886Unravelling some factors affecting sexual reproduction in rock-specialist shrub: insight from an endemic Daphne arbuscula (Thymelaeaceae)PLOS ONE

Dear Dr. Slovak,

Thank you for submitting your manuscript to PLOS ONE. After careful consideration, we feel that it has merit but does not fully meet PLOS ONE’s publication criteria as it currently stands. Therefore, we invite you to submit a revised version of the manuscript that addresses the points raised during the review process. Please submit your revised manuscript by Dec 22 2023 11:59PM. If you will need more time than this to complete your revisions, please reply to this message or contact the journal office at plosone@plos.org. Please include the following items when submitting your revised manuscript:A rebuttal letter that responds to each point raised by the academic editor and reviewer(s). You should upload this letter as a separate file labeled 'Response to Reviewers'.A marked-up copy of your manuscript that highlights changes made to the original version. You should upload this as a separate file labeled 'Revised Manuscript with Track Changes'.An unmarked version of your revised paper without tracked changes. You should upload this as a separate file labeled 'Manuscript'.

We look forward to receiving your revised manuscript.

Kind regards,

Harald Auge

Academic Editor

PLOS ONE

Journal Requirements:

4. We note that Supporting Figure S1 in your submission contain map/satellite images which may be copyrighted. All PLOS content is published under the Creative Commons Attribution License (CC BY 4.0), which means that the manuscript, images, and Supporting Information files will be freely available online, and any third party is permitted to access, download, copy, distribute, and use these materials in any way, even commercially, with proper attribution. For these reasons, we cannot publish previously copyrighted maps or satellite images created using proprietary data, such as Google software (Google Maps, Street View, and Earth). For more information, see our copyright guidelines: http://journals.plos.org/plosone/s/licenses-and-copyright.

(1) You may seek permission from the original copyright holder of Supporting Figure S1 to publish the content specifically under the CC BY 4.0 license.  

**Additional Editor Comments:**

Although your effort to sample data on plant reproduction in four local populations in the field was appreciated, one comment given by the reviewers refers to the relatively small sample size and low replication: based on four populations, reflecting a large range of population sizes, representing two different habitat types, and sampled in two years of different background climate, rigorous conclusions about the relative effects of these three factors can hardly be drawn. Hence, we recommend being more cautious when discussing implications that can be drawn from your results. Second, presentation of methods, results and discussion can be improved: some parts which are less informative for your main results can be moved to the supplementary material (e.g. the preliminary study), while others parts (e.g. results on breeding system) could be better elaborated. We also recommend restructuring the methods, results and discussion sections to improve the clarity of your manuscript: optimally, the different topics investigated should have the same order in all three sections, in line with the order how the objectives of your study are presented at the end of the introduction. Third, you may wish to place your case study in a broader context and include a brief discussion about the general relevance of your results for other species etc.

Reviewers' comments:

Reviewer's Responses to Questions

**Comments to the Author**

1. Is the manuscript technically sound, and do the data support the conclusions?

Reviewer #1: Partly

Reviewer #2: Partly

2. Has the statistical analysis been performed appropriately and rigorously? 

Reviewer #1: Yes

Reviewer #2: Yes

3. Have the authors made all data underlying the findings in their manuscript fully available?

Reviewer #1: Yes

Reviewer #2: Yes

4. Is the manuscript presented in an intelligible fashion and written in standard English?

Reviewer #1: Yes

Reviewer #2: Yes

5. Review Comments to the Author

Reviewer #1: In this manuscript, the authors studied flower and fruit production of four populations of the endemic Western Carpathian species Daphne arbuscular. The populations differed in their size and environmental conditions. In addition, the genetic background of the populations was investigated with the help of ddRad sequencing. The aim was to identify factors that could influence the fitness and long-term survival of this endangered species. The authors found that fruit production is generally low, below 50%, that cold conditions also have a negative effect on fitness and that sheltered microhabitats increased fruit set. Genetic factors may not critically influence the reproductive success of this endemic species. Overall, the study is educational, relevant, well written and (mostly) well designed and conducted. However, in the current version, I miss a more critical view of the results, as well as a discussion on the possibility of transferring them to other species and regions. Further suggestions for improvement are listed as "Minor comments".

Major comments:

1. I lack a critical view of the results. Only two years are considered, and yet inter-annual variability is examined. For reliable phenological studies (fruit set is in fact a phenological stage) longer time series would actually be needed. Furthermore, there is no replication between environmental conditions and population sizes. Conclusions such as those in lines 535-539 can therefore hardly be drawn. I know that both requirements cannot be fulfilled for this study and I appreciate the work that has been put into the data collection, but nevertheless these factsneed to be briefly discussed critically

2. The study is a case study with an endemic species in Slovakian mountains. In the discussion or conclusion, I would like to read something about whether and in which way the findings are relevant for other species and/or other regions. This would further increase the quality and relevance of the work.

Minor comments:

Funding Details: “Funders did not play any role tudy design, data collection and analysis, decision to publish, or preparation of the manuscript”. You probably wanted to write something like “study design?

L59: At this point I would like to read a bit more about the reasons for this. This is partly touched upon in the following, but 2-3 more sentences should be written here, as this is ultimately the core topic of this study.

L69-80: I would move some of these study specie’s properties to the M&M section and only keep the aspects which reasoning it as an “ideal model” (L86) for your research.

L138: You should introduce the abbreviations a few lines before ("Šiance (SI)") and add where the overview of the populations can be found.

L144: I am missing here that you have chosen individuals of the same size (a proxy for age?). I assume that the size of the studied individuals has a strong influence on your findings. If individuals of equal size were selected, this should be mentioned here. If not, this would have to be discussed.

L145: Of course, it is hard to tell but how have you defined an “ample size”?

L150-161: Scientifically, it is very accurate to do this preliminary study. But I think it is described too extensively. I have the opinion that this section can be deleted completely.

L176: Why were these inflorescences packed? To avoid the loss of flowers or fruits by wind? If I understand it correctly, an inflorescence could simply have been marked for this data recording. It should also be briefly mentioned in the discussion that this method may have affected the fruit set.

L187: Looking at the figure I can’t say which line is representing which microhabitat because all the lines a thinner. Please revise.

L201-275: The genetic methods are described very extensively. I would shorten the description for the main text and put the details in an appendix.

L303: Here an elsewhere. I stumbled a bit over the term "weather conditions". I think "climatic conditions" or "environmental conditions" is more appropriate for this study as I think of weather as short-term conditions of temperature and precipitation.

L338: You present the results of the interaction of time and population. This means that the populations show different changes between the years.

L404: Have you statistically tested the differences between the years? If not, I find the wording "significant" unfortunate.

L457: Since the flowering period is so short and can shift due to climate changes, mismatches can also be an important issue here.

L533: I would delete “[Gajdosova, work in progress]”.

Reviewer #2: To the Authors

The research presented in this manuscript will assist with conservation efforts for the endemic Daphne arbuscula. The manuscript focusses on three main topics: 1) reproductive output, population genetics, and role of climate/environment on reproductive output. In addition, the manuscript presents some new data on the breeding system of the species. Reproductive differences are related to microhabitat and environmental differences, but not associated with population genetics. Also, it was confirmed that the species is self-incompatible.

The introduction and discussion provide the necessary background and explanation of the results. However, the method and result sections will need some work to improve clarity. These sections have information that could be presented as part of supplements (i.e., 0.5 mm mesh size preliminary study, ITS and cpDNA work, weather data).

Comments and suggestions have been made to improve clarity and specific comments and questions are below and in the manuscript.

Abstract – Minor edits have been suggested, see manuscript for suggestions.

Methods – This section needs some improvement as some aspects of the study are not clear or well presented.

Breeding System: this section does not need all the text associated with lines 150-161 (mesh size 0.5 mm). Although it is useful to know that a preliminary study was conducted to determine the mesh size that could prevent thrips from pollinating the flowers, all this information can go in a supplement. In addition, it should be made clear that autogamy is being evaluated, then compatibility, see manuscript for suggestions.

Genetic Analyses: this section does not need the text associated with ITS and cpDNA, as this is distracting from the population genetic analyses. Again, this will be best as a supplement and mentioned in the discussion.

Results – This section needs some improvement as the presentation of results is not clear and some of the text could go in the methods and/or a supplement (lines 364-377).

The presentation of the Breeding System work is inadequate, and the heading of this section does not reflect the topic. The data presented in Supplement 2 should have been included in this section.

The text associated with ITS and cpDNA could be acknowledged in the discussion, as is not clear why this work was included as part of the manuscript.

Also, although the weather data is informative it is just descriptive and could be presented as a supplement. A lot of this information is repeated in the discussion.

Lastly, what is going on with VS29-Figure 4C?

Discussion – A few introductory sentences will help frame the discussion better. Overall, the discussion provides support for the results and addresses the factors that are driving fruit production directly (i.e., climate, breeding system) and indirectly (e.g., pollinators). However, it should be noted that the order of the topics in the discussion does not follow the way they were presented in the methods and results.

Citations – Some citations do not follow the journal’s format.

Figures and Tables – Figure 1 and 2 can be part of a supplement.

Supplemental Material – Supplements 1 and 2 should be included as part of the manuscript. Supplement 1 could be incorporated as part of Figure 4.

6. PLOS authors have the option to publish the peer review history of their article (what does this mean?). If published, this will include your full peer review and any attached files.

Reviewer #1: No

Reviewer #2: No

---

## [Author Response · Author response to Decision Letter 0]

25 Nov 2023

Editor Comments:

Although your effort to sample data on plant reproduction in four local populations in the field was appreciated, one comment given by the reviewers refers to the relatively small sample size and low replication: based on four populations, reflecting a large range of population sizes, representing two different habitat types, and sampled in two years of different background climate, rigorous conclusions about the relative effects of these three factors can hardly be drawn. Hence, we recommend being more cautious when discussing implications that can be drawn from your results. Second, presentation of methods, results and discussion can be improved: some parts which are less informative for your main results can be moved to the supplementary material (e.g. the preliminary study), while others parts (e.g. results on breeding system) could be better elaborated. We also recommend restructuring the methods, results and discussion sections to improve the clarity of your manuscript: optimally, the different topics investigated should have the same order in all three sections, in line with the order how the objectives of your study are presented at the end of the introduction. Third, you may wish to place your case study in a broader context and include a brief discussion about the general relevance of your results for other species etc.

Thank you for all the recommendations; we have accepted and incorporated each of them. Please find detailed responses to specific comments below.

Reviewer #1

In this manuscript, the authors studied flower and fruit production of four populations of the endemic Western Carpathian species Daphne arbuscular. The populations differed in their size and environmental conditions. In addition, the genetic background of the populations was investigated with the help of ddRad sequencing. The aim was to identify factors that could influence the fitness and long-term survival of this endangered species. The authors found that fruit production is generally low, below 50%, that cold conditions also have a negative effect on fitness and that sheltered microhabitats increased fruit set. Genetic factors may not critically influence the reproductive success of this endemic species. Overall, the study is educational, relevant, well written and (mostly) well designed and conducted. However, in the current version, I miss a more critical view of the results, as well as a discussion on the possibility of transferring them to other species and regions. Further suggestions for improvement are listed as "Minor comments".

Major comments:

1. I lack a critical view of the results. Only two years are considered, and yet inter-annual variability is examined. For reliable phenological studies (fruit set is in fact a phenological stage) longer time series would actually be needed. Furthermore, there is no replication between environmental conditions and population sizes. Conclusions such as those in lines 535-539 can therefore hardly be drawn. I know that both requirements cannot be fulfilled for this study and I appreciate the work that has been put into the data collection, but nevertheless these facts need to be briefly discussed critically.

Thanks for these comments. We agree with you that we should be more careful with some statements in discussion with respect to the impact of climate on fruit production, especially in light of our study design, which is unfortunately rectricted both with respect to number of replicates and zears of observation. Thanks for these comments. We agree with you that we should be more careful with some statements in discussion with respect to the impact of climate on fruit production, especially in light of our study design, which is restricted both with respect to the number of replicates and the years of observation. We acknowledge it in the discussion but also advocate why we designed it in this way

see p. 17-18 lns. 402-417:

Nevertheless, it is reasonable to approach our findings with caution due to some limitations in our study design. The lack of replication for each location, representing distinct climate types and population sizes, renders the design insufficiently robust, potentially strengthening the investigation's results. Adding more replications, however, presented considerable challenges, given the difficulty or inaccessibility of the majority of populations of this endemic shrub located on rocky cliffs with low population sizes. Similarly, the study's limitation to two seasons may be considered a constraint in drawing definitive conclusions. While we acknowledge that extending the study period to 10 or 15 years would provide sufficient data to discern clear trends in the impact of seasonal climatic conditions on fruit production, such an extended duration is evidently unfeasible for the majority of scientific projects today. Nevertheless, our observations across two seasons, encompassing climatically disparate years with divergent patterns during the reproductive phase of this species (Figure 2B and S1 File), highlight the influence of seasonal climate conditions on the fruit production of the studied species, even at the microscale level. Despite these constraints, we believe that these limitations have not unduly biased our outcomes or, particularly, the interpretation of our data.

We also improved sentences on p. 21 lines 472-483: 

Our analyses consistently demonstrated a decline in fruit production across all investigated populations, regardless of population size (cf. Table 1; S1 Table). This suggests that population numbers, particularly the number of mating partners within a population, may not play a decisive role in the reduction of the species' sexual reproduction. Several studies have emphasized the significance of population size and density for the reproductive success of animal-pollinated plants [reviewed by 101]. A comparable absence of correlation between sexual reproduction-associated traits and population size, as observed in D. arbuscula, was also identified in Allysum montanum from the Jura Mountains in France. [102]. Conversely, an opposite trend was detected for various plant species by [15]. Nevertheless, given the limitations in our sampling designs (see Discussion above), we exercise caution in making definitive statements on this matter. Further research is warranted to address this question comprehensively. 

2. The study is a case study with an endemic species in Slovakian mountains. In the discussion or conclusion, I would like to read something about whether and in which way the findings are relevant for other species and/or other regions. This would further increase the quality and relevance of the work.

Thank you once more for your highly pertinent comment. The following texts concerning the mentioned issue were added to the discussion section:

Seasonal climatic conditions were also identified as a key determinant of sexual reproduction success in high-mountain cushion saxicolous members of the genus Saxifraga [78, 79]. (p. 16-17 lns. 381-383)

The degree of fructification was found to depend on the temperature and occurrence of rather sunny and dry weather in spring also in glacial relict Pulsatilla vernalis [76]. (p. 18 lns. 421-423)

Indeed, late summer frosts exert a more pronounced negative impact on reproductive shoots, consequently affecting sexual reproduction success, compared to vegetative ones in the majority of studied high-mountain plants from the European Alps [82]. (p. 18 lns. 425-428) 

Similar microsite-dependent sexual reproduction success was observed in the high-alpine species Saxifraga moschata [78]. Plants from early snowmelt sites exhibited a higher abundance of sexual reproduction structures compared to those from later snowmelt sites. (p. 19 lns. 448-450)

A comparable absence of correlation between sexual reproduction-associated traits and population size, as observed in D. arbuscula, was also identified in Allysum montanum from the Jura Mountains in France. [102]. Conversely, an opposite trend was detected for various plant species by [15]. 

(p. 19 lns. 477-480)

Minor comments:

Funding Details: “Funders did not play any role tudy design, data collection and analysis, decision to publish, or preparation of the manuscript”. You probably wanted to write something like “study design?

Yes, you are right; it is only a typo. We corrected it.

L59: At this point I would like to read a bit more about the reasons for this. This is partly touched upon in the following, but 2-3 more sentences should be written here, as this is ultimately the core topic of this study.

We added the following statement to the Introduction chapter:

Endemics are frequently habitat specialists that have evolved more specific reproductive and pollination biology [11]. A decrease in lower seed production, for example, may be restricted by diminished pollen investment [4, 9], pollinator deficiency [12], or increased seed predation [13]. (p. 3 lns. 58-61)

L69-80: I would move some of these study specie’s properties to the M&M section and only keep the aspects which reasoning it as an “ideal model” (L86) for your research.

Done, moved to the Methods section. (p. 5-6 lns. 121-127 and p. 7 lns. 170-172)

L138: You should introduce the abbreviations a few lines before ("Šiance (SI; Fig 1A; S1 Fig) ") and add where the overview of the populations can be found.

We added an abbreviation for SI and a reference to the figure with a map (p. 6 lns. 143)

L144: I am missing here that you have chosen individuals of the same size (a proxy for age?). I assume that the size of the studied individuals has a strong influence on your findings. If individuals of equal size were selected, this should be mentioned here. If not, this would have to be discussed.

Yes, we have chosen adult individuals, which were approximately the same size. We also modified the following sentence in the “Sampling and design of the study” of the Material and Methods section:

Ten mature individuals of approximately the same size were selected from each population and marked with a unique metal identifier for subsequent identification purposes (S1 Table). (p. 7 lns. 150-151)

L145: Of course, it is hard to tell but how have you defined an “sample size”?

We explain this in the text as follows:

All the examined populations were of sufficient size to ensure that the research did not negatively impact their fitness and viability. Population size was estimated using data from the monitoring of D. arbuscula conducted by the Muránska planina National Park Administration. (p. 7 lns. 151-154)

L150-161: Scientifically, it is very accurate to do this preliminary study. But I think it is described too extensively. I have the opinion that this section can be deleted completely.

The paragraph about the preliminary study was moved to the supplementary materials (S2 File).

L176: Why were these inflorescences packed? To avoid the loss of flowers or fruits by wind? If I understand it correctly, an inflorescence could simply have been marked for this data recording. It should also be briefly mentioned in the discussion that this method may have affected the fruit set.

The shoots bearing flowers were primarily packed to prevent the loss of both formed and unformed fruits. Therefore, our method did not negatively impact the fruit set in our study. We have included the following explanation in the Material and Methods section:

The fruits typically ripen within 4-5 weeks; however, this process is highly individual and influenced by climatic conditions. Subsequently, the fruits disperse rapidly and randomly from shrubs within a few days. Therefore, after the majority of the flowering shoots on the studied individuals had been in the full-bloom stage for at least a few days, securing pollination, they were isolated using nylon bags to ensure precise analysis of the fruit set. (p. 7 lns. 162-166)

L187: Looking at the figure I can’t say which line is representing which microhabitat because all the lines a thinner. Please revise.

Done. We changed lines of microhabitats to dashed ones (S1 File).

L201-275: The genetic methods are described very extensively. I would shorten the description for the main text and put the details in an appendix.

The section, including parts about ITS and cpDNA amplification, was moved to the supplement (S3 File).

L303: Here an elsewhere. I stumbled a bit over the term "weather conditions". I think "climatic conditions" or "environmental conditions" is more appropriate for this study as I think of weather as short-term conditions of temperature and precipitation.

Changed word ‘wheather’ to ‘climatic’ wherever reasonable across the manuscript.

L338: You present the results of the interaction of time and population. This means that the populations show different changes between the years.

Corrected. p. 13 lns. 315-316: The populations showed significant differences in fruit production between the years

L404: Have you statistically tested the differences between the years? If not, I find the wording "significant" unfortunate.

No, differences in climatic conditions were not statistically evaluated. Corrected (and moved into S1 File following comments of Reviewer 2).

L457: Since the flowering period is so short and can shift due to climate changes, mismatches can also be an important issue here.

We agree with this statement. We added the following sentence to the discussion:

An additional critical factor adversely impacting successful pollination could be the mismatch in plant phenology and pollinator activity resulting from climate change [e.g., 84]. (p. 19, lns. 435–437)

L533: I would delete “[Gajdosova, work in progress]”.

Deleted.

Reviewer #2: To the Authors

The research presented in this manuscript will assist with conservation efforts for the endemic Daphne arbuscula. The manuscript focusses on three main topics: 1) reproductive output, population genetics, and role of climate/environment on reproductive output. In addition, the manuscript presents some new data on the breeding system of the species. Reproductive differences are related to microhabitat and environmental differences, but not associated with population genetics. Also, it was confirmed that the species is self-incompatible.

The introduction and discussion provide the necessary background and explanation of the results. However, the method and result sections will need some work to improve clarity. These sections have information that could be presented as part of supplements (i.e., 0.5 mm mesh size preliminary study, ITS and cpDNA work, weather data).

Comments and suggestions have been made to improve clarity and specific comments and questions are below and in the manuscript.

Thank you very much for your comments. We implemented all your specific comments and suggestions in the manuscript.

Abstract – Minor edits have been suggested, see manuscript for suggestions.

Corrected accordingly.

Methods – This section needs some improvement as some aspects of the study are not clear or well presented.

Breeding System: this section does not need all the text associated with lines 150-161 (mesh size 0.5 mm). Although it is useful to know that a preliminary study was conducted to determine the mesh size that could prevent thrips from pollinating the flowers, all this information can go in a supplement. In addition, it should be made clear that autogamy is being evaluated, then compatibility, see manuscript for suggestions.

The part about the preliminary study moved to the supplement (S2 File). Autogamy and compatibility were corrected according to suggestions in the manuscript.

 Genetic Analyses: this section does not need the text associated with ITS and cpDNA, as this is distracting from the population genetic analyses. Again, this will be best as a supplement and mentioned in the discussion.

The section about ITS and cpDNA amplification was moved to the supplement file (S3 File).

Results – This section needs some improvement as the presentation of results is not clear and some of the text could go in the methods and/or a supplement (lines 364-377).

Information about ITS and cpDNA amplification was moved to the supplement (S3 File). Sections including basic information about RADseq data (original lines 368–378)

---

## [Decision Letter · Decision Letter 1]

6 Feb 2024

PONE-D-23-32886R1Unravelling some factors affecting sexual reproduction in rock-specialist shrub: insight from an endemic Daphne arbuscula (Thymelaeaceae)PLOS ONE

Dear Dr. Slovak,

Thank you for submitting your revised manuscript to PLOS ONE. After careful consideration, we feel that it has merit but does not fully meet PLOS ONE’s publication criteria as it currently stands. Therefore, we invite you to submit a revised version of the manuscript that addresses the points raised during the review process.

There are only a few minor edits necessary. Reviewer #2 included their comments in the attached copy of the manuscript, while reviewer #1 suggests separating Fig 1A from the other panels of Fig 1 and move it to the Materials and Methods section. The section "Sampling and design of the study" is actually the place were you refer to the map of geographical locations, thus it would indeed be more useful to include Fig 1A here, while Fig 1B - D show genetic distances and structures of the populations and should thus remain in the Results section. In Fig 1A it seems to me, furthermore, that the geographical distance between populations SI and MS is missing. Once you will have considered these minor comments, I will most likely evaluate the revised manuscript by myself and will not request any further reviews.

We look forward to receiving your revised manuscript.

Kind regards,

Harald Auge

Academic Editor

PLOS ONE

Journal Requirements:

Reviewers' comments:

Reviewer's Responses to Questions

**Comments to the Author**

1. If the authors have adequately addressed your comments raised in a previous round of review and you feel that this manuscript is now acceptable for publication, you may indicate that here to bypass the “Comments to the Author” section, enter your conflict of interest statement in the “Confidential to Editor” section, and submit your "Accept" recommendation.

Reviewer #1: All comments have been addressed

Reviewer #2: All comments have been addressed

2. Is the manuscript technically sound, and do the data support the conclusions?

Reviewer #1: Yes

Reviewer #2: Yes

3. Has the statistical analysis been performed appropriately and rigorously? 

Reviewer #1: Yes

Reviewer #2: Yes

4. Have the authors made all data underlying the findings in their manuscript fully available?

Reviewer #1: Yes

Reviewer #2: Yes

5. Is the manuscript presented in an intelligible fashion and written in standard English?

Reviewer #1: Yes

Reviewer #2: Yes

6. Review Comments to the Author

Reviewer #1: I appreciate the great effort you have made to revise this manuscript. Below are some suggestions for improvement, which I think should be taken into consideration:

Introduction/Figure 1: It is somewhat confusing that the results of the molecular analyses are already shown in Figure 1. I would separate Figure 1A from the other panels. 1A is then shown as a separate figure in the Introduction and 1B-D in the results (because this a major part of your results and discussion!). Since the revision means that there are only two figures in the main text, I see no problem in showing three figures. Alternatively, Figure 1 could also be mentioned in the M&M. I would reformulate lines 96-100 e.g.: “In this study we investigated four populations of D. arbuscula, although limited to one mountain system (Muránska planina Mts in the Western Carpathians, Slovakia), demonstrate significant niche diversity, specifically those two residing at lower altitudes and facing south have typically drier and warmer climatic conditions, whereas those two populations located at higher altitudes on the northern slopes experience harsher and wetter conditions with a montane microclimate [19,20].”

L330: “populations” not “population”.

Reviewer #2: Dear Authors, thank you for addressing my comments. I only have a few minor edits that can be addressed during the proofing of the manuscript. It was nice to learn about your research and the conservation work associated with endemic Daphne arbuscula (Thymelaeaceae).

7. PLOS authors have the option to publish the peer review history of their article (what does this mean?). If published, this will include your full peer review and any attached files.

Reviewer #1: No

Reviewer #2: No

---

## [Author Response · Author response to Decision Letter 1]

17 Feb 2024

Journal Requirements:

We checked all the citations and the reference list.

Review Comments to the Author:

Reviewer #1: 

I appreciate the great effort you have made to revise this manuscript. Below are some suggestions for improvement, which I think should be taken into consideration:

Introduction/Figure 1: It is somewhat confusing that the results of the molecular analyses are already shown in Figure 1. I would separate Figure 1A from the other panels. 1A is then shown as a separate figure in the Introduction and 1B-D in the results (because this a major part of your results and discussion!). Since the revision means that there are only two figures in the main text, I see no problem in showing three figures. Alternatively, Figure 1 could also be mentioned in the M&M. I would reformulate lines 96–100, e.g.: “In this study we investigated four populations of D. arbuscula, although limited to one mountain system (Muránska planina Mts in the Western Carpathians, Slovakia), demonstrate significant niche diversity, specifically those two residing at lower altitudes and facing south have typically drier and warmer climatic conditions, whereas those two populations located at higher altitudes on the northern slopes experience harsher and wetter conditions with a montane microclimate [19,20].”

Thank you for your continued efforts in revising our manuscript. We appreciate your valuable proposals and recommendations. We have implemented the suggested changesas follows: 

1) We followed your suggestion and split Figure 1 into Figs. 1 and 3. Figure 1 (former Figure 1A) now includes only the distribution map and is placed in the Materials and Methods section. The formed Figures 1B-D, containing genetic analyses, have been relocated to the Results section and are now presented as Figs 3A-C

2) L330: “populations” not “population”.

Corrected (p. 14 lns. 323)

Reviewer #2: 

Dear Authors, thank you for addressing my comments. I only have a few minor edits that can be addressed during the proofing of the manuscript. It was nice to learn about your research and the conservation work associated with endemic Daphne arbuscula (Thymelaeaceae).

Thank you for your additional valuable comments and edits. We implemented them in the manuscript:

1) Following words were reworded accordingly: 

 p. 7 lns. 157-159 and 172; 

 p. 9 lns. 207, 210, 214, 219; 

 p. 13 lns. 310, 313; 

p. 19 lns. 434

2) Deleted heading “Statistical analysis” in M&M section

Deleted.

3) This is not a heading, is a statement Perhaps change to "Role of genetic variability on sexual reproduction"

Reworded heading in the Discussion section: Done as recommended (p. 21 lns. 495-496): „Low association between genetic variability at the population level and sexual reproduction in D. arbuscula“ 

4) Correction in the financial disclosure (typo) - updated statement:

Done.

---

## [Editor Report · Decision Letter 2]

6 Mar 2024

Unravelling some factors affecting sexual reproduction in rock-specialist shrub: insight from an endemic Daphne arbuscula (Thymelaeaceae)

PONE-D-23-32886R2

Dear Dr. Slovak,

We’re pleased to inform you that your manuscript has been judged scientifically suitable for publication and will be formally accepted for publication once it meets all outstanding technical requirements.

Kind regards,

Harald Auge

Academic Editor

PLOS ONE